# Are LCA Studies on Bulk Mineral Waste Management Suitable for Decision Support? A Critical Review

Christian Dierks [1,*], Tabea Hagedorn [1], Alessio Campitelli [1], Winfried Bulach [2] and Vanessa Zeller [1]

1 Chair of Material Flow Management and Resource Economy, Institute IWAR, Technische Universität Darmstadt, Franziska-Braun-Strasse 7, 64287 Darmstadt, Germany; t.hagedorn@iwar.tu-darmstadt.de (T.H.); a.campitelli@iwar.tu-darmstadt.de (A.C.); v.zeller@iwar.tu-darmstadt.de (V.Z.)
2 Oeko-Institut e.V., Rheinstrasse 95, 64295 Darmstadt, Germany; w.bulach@oeko.de
* Correspondence: c.dierks@iwar.tu-darmstadt.de

**Abstract:** Bulk mineral waste materials are one of the largest waste streams worldwide and their management systems can differ greatly depending on regional conditions. Due to this variation, the decision-making context is of particular importance when studying environmental impacts of mineral waste management systems with life cycle assessment (LCA). We follow the premise that LCA results—if applied in practice—are always used in an improvement (i.e., decision-making) context. But how suitable are existing LCA studies on bulk mineral waste management for decision support? To answer this question, we quantitatively and qualitatively assess 57 peer-reviewed bulk mineral waste management LCA studies against 47 criteria. The results show inadequacies regarding decision support along all LCA phases. Common shortcomings are insufficient attention to the specific decision-making context, lack of a consequential perspective, liberal use of allocation and limited justification thereof, missing justifications for excluded impact categories, inadequately discussed limitations, and incomplete documentation. We identified the following significant issues for bulk mineral waste management systems: transportation, the potential leaching of heavy metals, second-order substitution effects, and the choice to include or exclude avoided landfilling and embodied impacts. When applicable, we provide recommendations for improvement and point to best practice examples.

**Keywords:** life cycle assessment; LCA; mineral waste management; CDW; slag; decision support; critical review

## 1. Introduction

The management of bulk mineral waste is associated with relatively low environmental impacts when compared to the respective producing systems such as steel production or the whole life cycle of a building. Nonetheless, aggregate recycling can contribute to reducing impacts in comparison to natural aggregate production and waste material landfilling. This can be significant in absolute terms, because bulk mineral waste materials such as construction and demolition waste (CDW), asphalt waste, and metallurgical slags are one of the largest waste streams worldwide. Over 357 million tons of non-hazardous mineral waste from construction and demolition was produced in the European Union in 2018, with Germany, United Kingdom, France, and Italy as the largest producers [1]. In the same year, the United States produced 405 million tons of concrete waste alone, almost 20% of which was landfilled [2]. China produced 1.13 billion tons of CDW in 2014 [3]. At the same time, recycled aggregates are viable substitutes for virgin mineral resources such as limestone, gravel, and sand in several applications, most notably road construction (bound and unbound) as well as concrete. The European Union Directive 2008/98/EC (Waste Framework Directive) set the target that 70% of non-hazardous CDW shall be recovered by the end of 2020. Indeed, high recovery rates have been achieved in many European countries. However, this recovery predominantly comprises open-loop recycling

into inferior applications such as backfilling and the recovery as low-quality aggregates in road sub-bases, reducing the intrinsic quality of the material [4]. Similarly, the recycling of asphalt waste (replacing virgin asphalt or aggregate) and metallurgical slags (replacing binder or aggregate) is well established [5,6].

Mineral waste management networks are complex systems [7]. Depending on regional conditions, they can differ greatly, e.g., in terms of involved stakeholders [7,8], transport distances [9], demand for recycled aggregates, and type of application [10]. It is therefore not self-evident that replacing virgin with secondary materials will in fact lead to a reduction in environmental impacts. This means that decision-making with respect to mineral waste management systems needs to be supported by case-specific information.

Life cycle assessment (LCA), which is standardized in ISO 14040/44 [11,12], serves as a decision support tool that enables practitioners to investigate if and to what extent environmental impacts increase or decrease by choosing different courses of action in a given decision-making context.

ISO 14040 asserts in Annex A.2 that "[ . . . ] the products and processes studied in an LCA are those affected by the decision that the LCA intends to support" and stresses that "it is necessary to consider the decision-making context when defining the scope of an LCA". It further explains that any LCA is ultimately used in an improvement context whenever the information is applied in practice [11]. This is further supported by the four examples given in ISO 14040 for direct applications of LCA results shown in Figure 1. Product development and improvement, strategic planning, and public policy-making are all direct decision-making situations. The use of LCA results in marketing is a form of decision support for consumers and policy-makers, intended to direct their choices. We agree with this point and argue that LCA is therefore always—directly or indirectly— a decision support tool and has to be treated as such by the practitioner. Under this premise, accounting contexts without decision support, as defined for example in the ILCD handbook [13], do not apply.

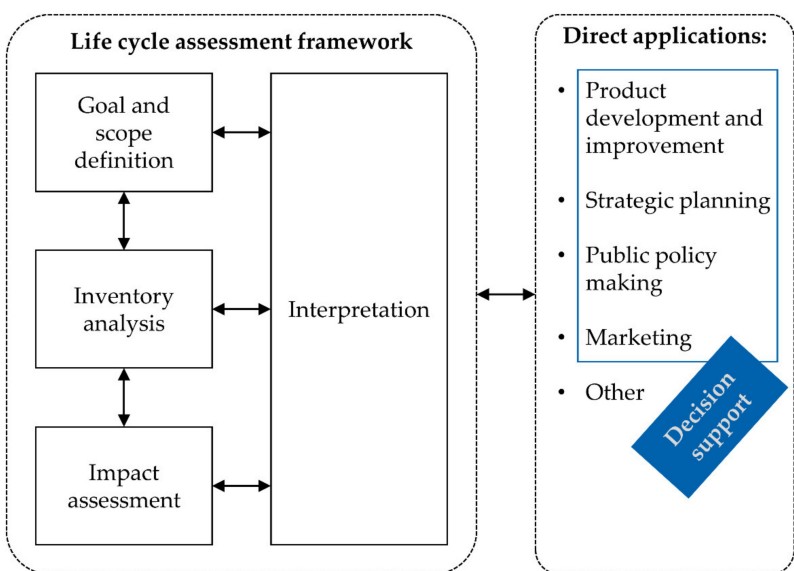

**Figure 1.** Life cycle assessment framework and direct applications based on [11,12].

Divergent or contradictory LCA results can arise due to methodological variations, e.g., choice of functional unit, modeling approach, impact categories, and characterization models. However, they can also be a consequence of real-world differences between the systems investigated by different studies, e.g., temporal and geographical characteristics [14] or the quantitative scope. These real-world differences can be explained by the fact that the decision-making contexts can vary vastly between LCA studies. This is especially true for mineral waste management systems, where several types of decisions can be supported by LCA. Public

policy-making can affect waste management systems at municipal, regional (e.g., [10]), or national level (e.g., the planned German secondary building materials directive). The choice of primary versus secondary aggregate for concrete mixes or road sub-bases in public or private procurement may have implications for individual construction projects (e.g., [15]) or for a larger geographic and temporal context (e.g., [16]). Marketing of secondary construction materials can be addressed at customers and policy-makers at the municipal, regional, national, or even international level. To deliver robust decision support using LCA, it is necessary to consider these real-world differences, which are a natural result of the specific decision-making context.

Consequently, we intend to answer the following research question: are the existing LCA case studies on bulk mineral waste management suitable for decision support? The associated goal of this review is to identify methodological and data-related challenges with special emphasis on suitability for decision support. To this end, we assess 57 peer-reviewed journal articles against 47 criteria, discuss potential shortcomings, and provide recommendations.

To the best of our knowledge, no critical review of LCA studies on bulk mineral waste management with focus on decision support has been conducted before. A number of existing LCA reviews focus on the use of mineral waste materials in product systems, i.e., concrete [17–20], cement [21], and highway pavements [22]. Other reviews focus on CDW management in particular [23,24] or on circular economy in the construction and demolition sector [9]. Laurent et al. [25,26] conducted an extensive two-part review of over 200 LCA studies in the context of waste management as a whole and dissected the body of literature regarding its methodology and shortcomings thereof. However, they focused on waste management as a whole (as opposed to addressing specific waste fractions) and reviewed the general LCA methodology without focus on decision support. Furthermore, just six of the 57 studies assessed in the present review were featured in Laurent et al. [25,26]. Where applicable, we discuss the conclusions of Laurent et al. [26] in the context of this article.

## 2. Materials and Methods

Literature search, eligibility screening process, and data extraction were conducted according to the PRISMA guidelines for systematic reviews [27]. The completed PRISMA checklist based on Moher et al. [28] is available in the Supplementary Materials (Table S1).

### 2.1. Search Strategy and Screening Process

The literature search and screening process followed the steps outlined in Figure 2. Scopus and Web of Science were used to conduct the search, filtering by keywords to identify relevant waste fractions (e.g., 'mineral waste', 'construction and demolition waste', 'slag') and methodology (e.g., 'life cycle assessment', 'life cycle analysis') in the title, abstract, and keywords. The complete search strings for both databases as well as refinements regarding document type, language, timespan, and science categories (Web of Science) or subject area (Scopus) are documented in the Supplementary Materials (Table S2); 2823 and 3715 records were identified through Web of Science and Scopus, respectively. Three additional records were identified by manually screening references of identified studies in a non-systematic way.

The eligibility screening process was performed manually by the first author using the reference management software Citavi 6.8. During the import into Citavi, 1934 duplicates were removed, leaving 4607 records to be filtered for eligibility. Studies had to fulfill all of the following eligibility criteria to be included in the quantitative and qualitative assessment:

- Methodology: Life Cycle Assessment.
- Type of study: Case study.
- Waste fraction: Non-hazardous bulk mineral waste materials.
- Perspective: Waste management.
- Publication type: Peer reviewed journal article.
- Language: English.

- Date of publication: After 2000 and before 15 February 2021.

**Figure 2.** Flowchart of the literature search and screening process to identify relevant publications for review based on PRISMA [28].

The scope of this review includes LCA case studies. We consider the ISO standards 14040/44 to be valid for all LCA studies and therefore did not limit the body of literature to studies explicitly claiming to follow the ISO standards. This choice is further supported by the fact that we do not assess the body of literature regarding its conformity with the ISO standards, but merely use ISO requirements as assessment criteria where applicable, as elaborated in the following section.

The waste fractions covered by this review are non-hazardous bulk mineral waste materials such as the mineral fraction of CDW, metallurgical slags, or asphalt waste. In this article, we define 'non-hazardous' as having a concentration or leaching potential of toxic elements below the legal threshold for use in the intended application, meaning that valorization of the waste material without elaborate pollutant removal is in fact a feasible option. Leaching and legal thresholds are only marginally addressed in the body of literature. For this reason—and only for the purpose of selecting studies eligible for review—we considered the fact that valorization is addressed in a study as the assumption by the authors that it is in fact legally feasible. This does not mean that the waste materials are free of toxic elements (see Section 3.1.4 for further discussion of leaching). Studies focusing on non-mineral fractions of CDW (e.g., [29]) or hazardous mineral waste such as tar-containing asphalt waste (e.g., [30]) were excluded.

The body of literature can be divided into two kinds of studies: one group follows the perspective of waste management, assessing valorization and disposal options for the waste material under investigation; the second group focuses on the consuming system, assessing, e.g., material compositions of concrete incorporating secondary materials. We only included the former group (what should we do with the waste?) and excluded the latter (which input materials should we choose?). Further, studies focusing predominantly on the producing system (e.g., building demolition, road remediation, metals production) were excluded.

Lastly, only peer-reviewed journal articles were included, as we assume that they best reflect the state of research and strike the right balance of comprehensiveness and quality assurance. Conference papers were excluded, as they are usually too short to contain all necessary

information. Grey literature such as technical reports and dissertations were excluded, because we assumed that their results would be published as journal articles if the authors considered the content to be state of the research. Only articles in English were included.

Regarding the temporal scope of this review, we include studies published after 2000, because ISO 14043, the last of the original ISO standards, was published in that year. Earlier studies cannot be judged fairly based on ISO requirements. We include studies published before the release of the revised 14040/44 standards in 2006, because the overall content of the standards remains largely unchanged [31].

According to these eligibility criteria, 4464 records were excluded by screening all records twice based on title, keywords, and abstract. The remaining 143 full texts were screened during the in-depth review. In this step, 86 studies were excluded, leaving 57 peer-reviewed journal articles to be included in the final review (see Appendix A Table A1).

The waste fractions CDW, metallurgical slags, and asphalt waste are investigated by 47, seven, and three studies, respectively. No studies focusing on the management of excavated earth were identified. The most common valorization routes are the use as aggregate (45 studies)—substituting limestone and other rocks, gravel, and/or sand—and as a binder in concrete or asphalt (11 studies)—replacing clinker, Portland cement, bitumen, or fly ash. The temporal distribution of identified publications is shown in Figure 3.

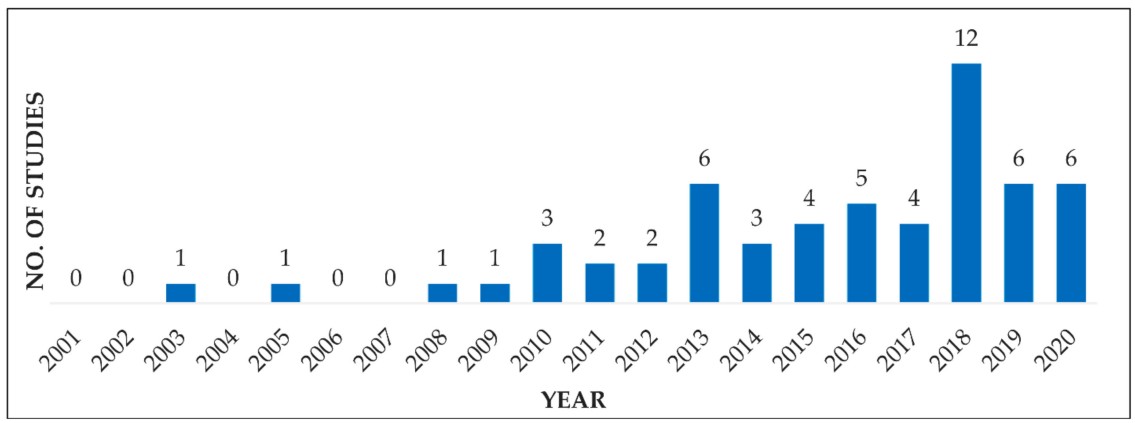

**Figure 3.** Temporal distribution of publications included in this review.

### 2.2. Assessment Procedure and Data Extraction

The identified studies were assessed regarding methodology and documentation with special emphasis on suitability for decision support. To this end, we defined 47 assessment criteria, which are summarized in Table 1. The assessment criteria cover all LCA phases: goal and scope definition, life cycle inventory analysis (LCI), life cycle impact assessment (LCIA), and interpretation. Yes–no questions were posed for a binary quantitative assessment of the literature, supplemented by a qualitative assessment where appropriate. Many of the assessment criteria relate to good LCA practice and documentation, as these are crucial for robust decision support. We chose requirements for general LCA practice defined in ISO 14040 and ISO 14044, which we deemed especially relevant for decision support. We further established criteria specific to LCA applied to mineral waste management based on significant issues identified in the body of reviewed literature and LCA case studies focusing on systems consuming mineral waste as secondary input material (e.g., concrete production). Lastly, we developed assessment criteria specifically for decision support in LCA, which are derived from the body of literature, our interpretation of the ISO standards (especially Annex A.2 of ISO 14040), and our own experience. For improved readability, the rationale for each criterion is discussed in cohesion with the assessment results in the results and discussion section. Data were extracted by a keyword search and manual screening of each full text and Supplementary Materials. Extracted data were managed in

a spreadsheet. The search terms used for data extraction as well as the yes–no conditions for the quantitative assessment are available in the Supplementary Materials (Table S3).

**Table 1.** Assessment criteria, specificity, and rationale for their inclusion (DS = decision support; MWM = mineral waste management).

| Category | Criterion | Specific for | Rationale |
|---|---|---|---|
| **Goal definition** | Is a goal defined? | LCA | ISO requirement |
| | Is the intended application declared? | LCA | ISO requirement |
| | Is the intended audience stated? | LCA | ISO requirement |
| | Is decision support identified as a goal? | DS | Derived from ISO |
| | Is the supported decision identified? | DS | Derived from ISO |
| | Is the decision-maker identified? | DS | Expert judgement |
| | Is the temporal scope of the decision identified? | DS | Expert judgement |
| | Is the spatial scope of the decision identified? | DS | Expert judgement |
| | Is the quantitative scope of the decision identified? | DS | Expert judgement |
| **Functional unit** | Is a function defined? | LCA | ISO requirement |
| | Is a functional unit defined? | LCA | ISO requirement |
| | Does the functional unit contain the function? | LCA | ISO requirement |
| | Is a reference flow defined? | LCA | ISO requirement |
| | Is the mineral waste composition defined? | MWM | Body of literature |
| | Are technical properties of the mineral waste material defined? | MWM | Body of literature |
| **Multifunctionality** | Is the study declared as consequential LCA? | DS | Expert judgement |
| | Are marginal supplying technologies identified? | DS | Expert judgement |
| | Is allocation avoided in the foreground system? | LCA | ISO requirement |
| | Is the general approach to multifunctionality stated? | LCA | ISO requirement |
| | Is the approach to multifunctionality justified? | LCA | ISO requirement |
| | Is a sensitivity analysis conducted on allocation? | LCA | ISO requirement |
| **Life cycle phases** | Are embodied impacts either excluded or justified? | MWM | Expert judgement |
| | Is the inclusion/exclusion of (avoided) landfilling justified? | MWM | Expert judgement |
| | Is the material processing included? | MWM | Body of literature |
| | Is transport included? | MWM | Body of literature |
| | Is leaching included? | MWM | Body of literature |
| **Inventory analysis** | Are foreground inventory data provided? | LCA | ISO requirement |
| | Is the background database stated? | LCA | Derived from ISO |
| | Is the background database version stated? | LCA | Derived from ISO |
| | Are the used datasets stated? | LCA | Derived from ISO |
| | Is the system model choice in ecoinvent 3 documented? | LCA | Derived from ISO |
| | Is the system model choice in ecoinvent 3 justified? | LCA | Derived from ISO |
| | Are technical parameters for substitution defined? | MWM | Body of literature |
| | Is a substitution factor used? | MWM | Body of literature |
| | Are other substitution effects identified and quantified? | MWM | Body of literature |
| **Impact assessment** | Is the LCIA methodology stated? | LCA | ISO requirement |
| | Are choices regarding LCIA methodology justified? | LCA | ISO requirement |
| | Are unweighted LCIA results provided? | LCA | ISO requirement |
| | Is disregarding impact categories justified? | LCA | ISO requirement |
| **Interpretation** | Are significant issues identified? | LCA | ISO requirement |
| | Is a sensitivity analysis performed? | LCA | ISO requirement |
| | Is a sensitivity check documented? | LCA | ISO requirement |
| | Is a completeness check documented? | LCA | ISO requirement |
| | Is a consistency check documented? | LCA | ISO requirement |
| | Are limitations discussed? | LCA | ISO requirement |
| | Are conclusions drawn? | LCA | ISO requirement |
| | Are recommendations provided? | LCA | ISO requirement |

The results are presented in aggregated form to prevent singling out studies, as the goal is to assess the identified body of literature as a whole and not any particular study. As Laurent et al. [26] point out, this disregards the development of LCA competence over

time. However, as visualized in Figure 3, the vast majority of studies were published in the last decade, which leads us to believe that this effect will not significantly affect our conclusions. Where applicable, we point out best practice examples.

## 3. Results and Discussion

This section follows the four phases of LCA defined in the ISO 14040-series [11], i.e., goal and scope definition (Section 3.1), inventory analysis (Section 3.2), impact assessment (Section 3.3), and interpretation (Section 3.4). It becomes apparent that most methodological choices are part of the goal and scope phase. Nevertheless, we allocated some criteria to the affected phase for improved readability (e.g., choice of impact categories to the impact assessment phase). Quantitative assessment results are expressed as percentages of the identified body of literature (57 studies) unless otherwise specified.

### 3.1. Goal and Scope Definition

Due to the large number of assessment criteria referring to the goal and scope definition, this section is subdivided into goal definition (Section 3.1.1), functional unit (Section 3.1.2), handling of multifunctionality (Section 3.1.3), and included life cycle phases (Section 3.1.4).

### 3.1.1. Goal Definition

The goal definition phase of LCA is of utmost importance, as it determines the exact approach to be followed [32]. All methodological choices, e.g., definition of the functional unit, system boundaries, allocation approach, and impact assessment method, depend on the goal of the study and the scientific question to be answered. Therefore, any third-party LCA report needs to document the reasons for carrying out the study, the intended applications, as well as the target audiences [12]. We agree with Laurent et al. [26] that in order to prevent misinterpretation, LCA case studies, including those mainly intended to support methodological development, should provide "sufficient information on the context of the study" (e.g., if and how the LCA results can be applied in practice).

Figure 4 shows that 95% of the LCA studies under investigation define a goal in the introduction or the goal and scope section, but just 46% declare the intended application, and 26% report the intended audience anywhere in the article or the Supplementary Materials. Furthermore, we can reaffirm the findings of Laurent et al. [26] regarding the goal definition phase, i.e., that the majority of studies merely write what they did, rather than why they did it. The most common type of goal is evaluating the potential impacts of a process or a waste material, with little or no additional information given about the intended application of the results. If documented, the intended application and audience are frequently included in the discussion or the conclusion instead of the goal and scope.

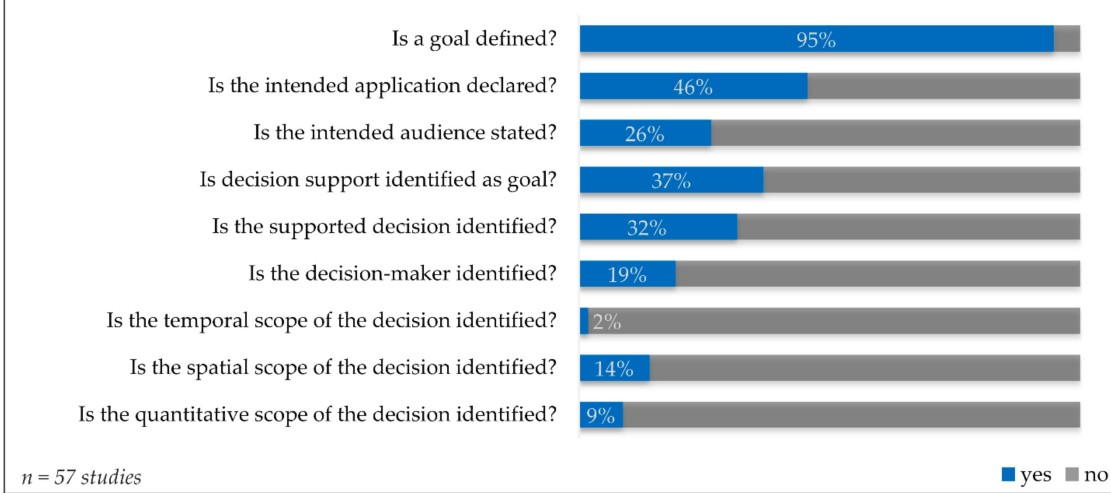

**Figure 4.** Assessment results for the category 'goal definition'.

This may indicate that these crucial issues are often little more than an afterthought for authors. As Schrijvers et al. [33] demonstrate, a broadly defined goal such as investigating "what is the environmental impact of a product" can lead to 15 or more different research questions and at least five different modeling approaches. This is problematic, because it can lead to inadequate methodological choices when conducting an LCA, as well as to misinterpretations when LCA results are applied in practice. The goal of the study, the direct applications, and the intended audience should hence be clearly defined within the goal and scope section.

As argued above, LCA results are always intended or likely to be used in specific decision-making contexts. It is therefore crucial to identify decision support as a goal of any LCA case study and consider the specific decision-making context as specified in Annex A.2 of ISO 14040 [11]. We argue that in order to evaluate the decision-making context, it is helpful to identify the decision-maker. This is especially important for mineral waste management systems, which can differ greatly in terms of involved stakeholders [7,8]. Identifying the decision-maker can assist in understanding the exact decision-making context, including its temporal, spatial, and quantitative scope. As asserted by Yang [34], the scale of change determines the degree to which other industries are affected, e.g., whether they will respond to an increase in demand using existing or newly created capacities. Divergent LCA results can be partially explained by real-world differences such as variations in materials under investigation as well as temporal, geographical, and technological characteristics. It is entirely possible that, given a certain place and time, the recycling of one material is environmentally feasible while that of another is not—or that the recycling of a given waste material is environmentally feasible in one location but not in another. In this regard, the geographical and temporal system boundaries can significantly affect several parameters such as transport distances, means of transport, technologies used, electricity and fuel mixes, as well as other background processes. Identifying the scale and the geographical and temporal scope of the decision is therefore essential for the correct definition of the system boundaries and the identification of representative inventory data.

Of the total case studies, 37% report decision support as the goal and 32% identify decisions intended to be supported by the LCA results; 19% of studies mention the decision-maker within their goal definition. However, studies often only vaguely define the identified decisions (e.g., "improving actions", "waste management strategies") and the decision-maker (e.g., "regional authorities", "local government", "policy", "waste producer"). The temporal, spatial, and quantitative scope of the decision at hand are reported by 2%, 14%, and 9%, respectively. Interestingly, the temporal (23%) and geographical (89%) foreground system boundaries are more commonly documented.

The common absence of decision support in the goal definitions indicates a lack of attention to the decision-making context in which the LCA results are intended or likely to be used. To avoid misinterpretation, the decision-making context, including the temporal, spatial, and quantitative scope of the decision, requires more attention by LCA practitioners. The geographical and temporal system boundaries are part of the scope definition and should be carefully chosen in line with the temporal and geographical scope of the decision at hand. Consequently, the scope of the decision has to be identified as part of the LCA goal definition. The criteria assessed in this section are crucial for all methodological choices in any given LCA and should be considered and reported by LCA practitioners.

### 3.1.2. Functional Unit

The functional unit represents the quantification of the functions of the systems under investigation [11]. It is of great importance in any LCA, because it serves as the basis for comparison between different systems and for further methodological choices such as the definition of system boundaries. Thus, it is necessary to identify—and report—the main common function(s) of the systems under investigation as well as to clearly define the functional unit as the quantification of said function(s) [12]. Furthermore, it is important to

define the reference flows, i.e., the type and amount of physical flows needed to provide the function defined in the functional unit, as they may vary between the different systems under investigation [11]. The technical properties of the waste material may be relevant to determine to what degree they can replace virgin materials and what possible trade-offs are to be expected. They can further influence the leaching behavior of the material, which is especially relevant when utilized as unbound aggregate. Bulk density is relevant for the estimation of environmental impacts of material transport.

The assessment results regarding the functional unit are summarized in Figure 5. Although 72% of studies define a functional unit, just 44% of studies identify the specific functions on the basis of which the systems are compared and 40% include them in the functional unit; 11% of studies explicitly report reference flows; 81% of studies report the material composition of the mineral waste fraction under investigation, i.e., the specific waste materials contained in the waste fraction. However, this includes all studies that investigate single waste materials such as concrete or blast furnace slag. Eleven out of the 35 studies investigating a CDW fraction containing multiple mineral waste materials such as concrete, bricks, or tiles did not clearly report the waste composition. Few studies report the chemical composition of the material. Technical properties of the waste material under investigation are defined by 23% of studies. Material density is the most commonly defined property (nine studies). Leaching properties of the waste material and aggregate qualities (low–mid–high) are defined by two studies each.

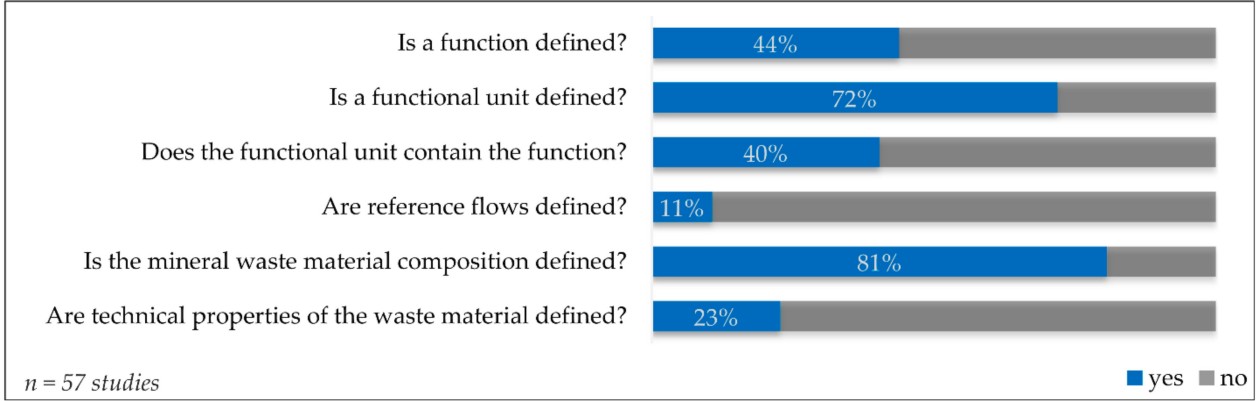

**Figure 5.** Assessment results for the category 'functional unit'.

The assessment results indicate that the choice of an appropriate functional unit requires more attention in practice. We advise accounting for all relevant functions of all systems within the functional unit. If there is no function included in the functional unit, it is for all intents and purposes a reference flow. We recommend accounting for the waste material composition as well as technical parameters of the waste material, as omitting these factors may lead to false assumptions and invalid comparisons, which can affect all subsequent phases of the LCA (e.g., substitution, transport, and leaching). If these parameters are unknown to the authors (e.g., because the LCA has a broad scope), this data gap should be discussed as a potential limitation of the study.

### 3.1.3. Multifunctionality

Attributional and consequential modeling are the main approaches used for system modeling in LCA. To date, no universally valid definitions exist for attributional LCA (ALCA) and consequential LCA (CLCA). In this review, we follow the definitions by Sonnemann and Vigon [35]:

- **Attributional approach:** "system modeling approach in which inputs and outputs are attributed to the functional unit of a product system by linking and/or partitioning the unit processes of the system according to a normative rule."

- **Consequential approach:** "system modeling approach in which activities in a product system are linked so that activities are included in the product system to the extent that they are expected to change as a consequence of a change in demand for the functional unit."

Kua [36] shows that the modeling approach, more precisely the choice between ALCA and CLCA, can influence the results significantly. He demonstrates in a case study that replacing sand with steel slag can be regarded as worse (ALCA) or potentially better (CLCA), depending on the selected approach.

We agree with Yang [34] that in order to support decision-making with LCA, the counterfactual needs to be estimated. Robust decision-making requires a good understanding of both the relevant facts (what will happen as a consequence of the decision?) and the underlying values (what do we want to happen as a consequence of the decision?). We argue that LCA results should contain as little normative influence as possible, to allow decision-makers to transparently merge them with the specific values they represent. The inclusion of values in form of normative rules for defining system boundaries is dangerous for two reasons:

- It risks merging different sets of values and therefore blurring the lines between different value systems (e.g., those of LCA practitioner and decision-maker).
- Even if the decision-maker's values are met by the chosen normative partitioning rule, the cart is still put in front of the horse: the best knowledge should lead decision-making, not the other way around.

This leads to the conclusion that only a consequential modeling approach representative of the specific decision-making context is suitable for decision support. A one-size-fits-all attributional approach cannot effectively support concrete decisions, as ALCA—by definition—does not keep the cause–effect principle intact. This assertion is supported by Weidema and Schmidt [37], who explain that mass and energy balances are broken by allocation. Consequential inventory modeling should include marginal suppliers, as they may be very different from the average market mix, in which case using the latter would introduce significant errors into the model [38].

According to ISO 14044 [12], the allocation procedure shall be clearly documented and justified, and whenever possible, subdivision or system expansion shall be applied to avoid allocation. ISO 14044 [12] further requires that a sensitivity analysis is conducted if alternative allocation procedures are applicable.

As visualized in Figure 6, only three studies (5%) declare having followed a consequential modeling approach. Marginal technologies affected by the change are clearly identified by two studies (4%). Twelve percent of studies report to have avoided allocation by using system expansion, while in 88% allocation is either employed or not addressed at all. Including the 12% of studies that report to have avoided allocation, 28% state their approach to handle multifunctionality in the foreground system and 21% justify their choices regarding allocation procedures as required by ISO 14044. It is worth noting that some studies explain and even justify the allocation procedure used for specific multifunctional processes but do not provide information on the general allocation procedure. Sensitivity analyses regarding allocation are found in three studies (5%).

More CLCA studies are needed to understand the implications of decisions in the bulk mineral waste management sector. We therefore recommend using a consequential modeling approach, including the use of marginal (as opposed to average) data, whenever LCA results are intended or likely to be used in a decision-making context. We further recommend clearly acknowledging that the results are not suitable for decision support if an attributional modeling approach is followed. Note that we thereby contradict the ILCD handbook's recommendation to use attributional modeling for micro-level decision support (Situation A) [13]. In order to keep mass and energy balances intact (which is required to represent the cause-effect chain), we recommend avoiding allocation by including processes in the system boundaries to the extent that they are expected to change as a consequence of the decision at hand.

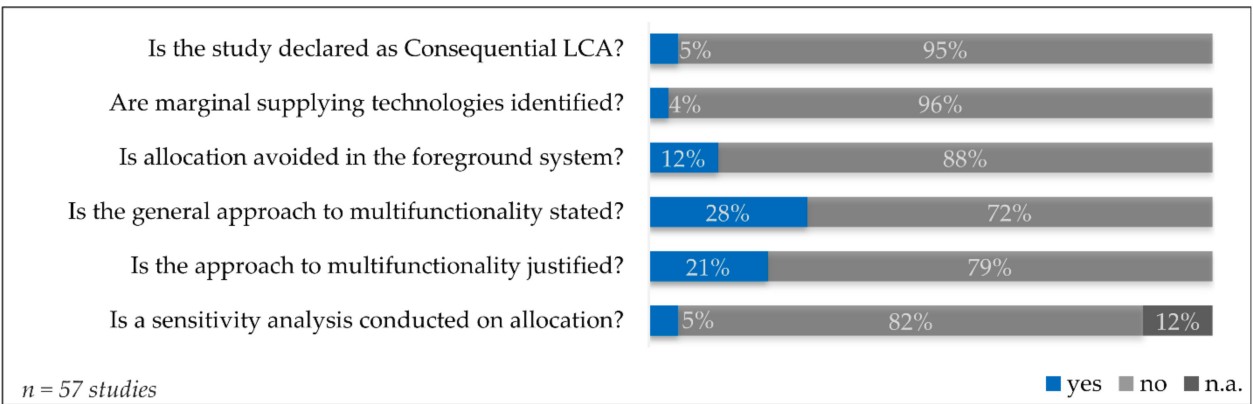

**Figure 6.** Assessment results for the category 'multifunctionality' (n.a. = not applicable: allocation was avoided).

### 3.1.4. Life Cycle Phases

Recycling systems are a special case in LCA, as they are inherently multifunctional, serving as both waste management and secondary resource production. They are therefore always located between upstream waste producing systems and downstream secondary material consuming systems. Regarding the selection of system boundaries, three points stand out: (1) life cycle phases of the upstream waste producing systems ('embodied impacts'), (2) potentially avoided landfilling, and (3) downstream life cycle phases. Regarding the latter, transportation and leaching of heavy metals were identified as potential significant issues.

When assessing different waste management options, processes of the waste producing system (e.g., a building or steel production) should be included in the system boundaries to the degree that they are affected by the decision at hand. The sunk cost fallacy [39], i.e., basing decisions on previous investments instead of future consequences, should be avoided by excluding **embodied impacts** not affected by the decision. **Landfilling** can contribute significantly to the LCIA results [10,16,40–42] and is a potential significant issue for LCAs on mineral waste management [10,41]. The choice to include or exclude avoided landfilling in/from the system boundaries should be based on the systems investigated and how they differ from one another. Landfilling can be addressed by including it in the system boundaries or by subtracting its impacts from systems where avoided landfilling is an additional function. ISO 14044 requires the omission of additional functions in functional units to be explained and reported [12].

The assessment results for these criteria are summarized in Figure 7. In 96% of studies, impacts from life cycle phases of the waste producing system (embodied impacts) are either excluded from the system boundaries or the reasons for their inclusion are explained; 68% of studies justify their choice to include or exclude landfilling either as a credit or as a burden.

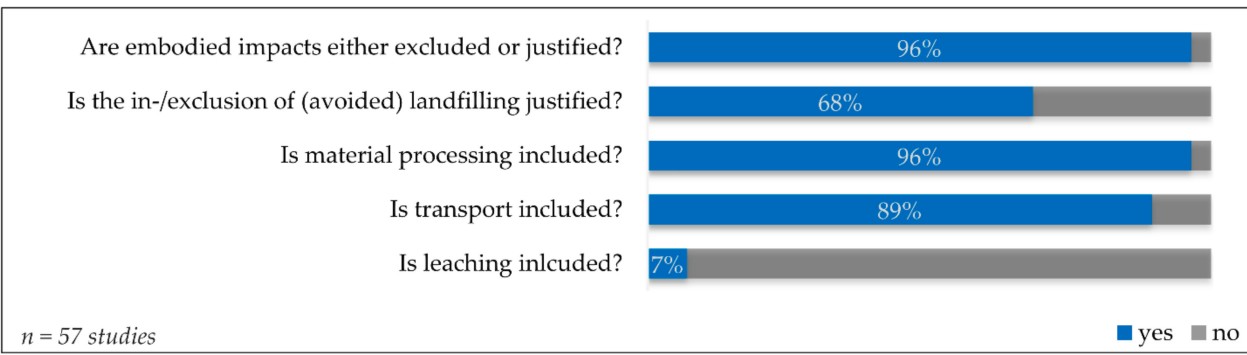

**Figure 7.** Assessment results for the category 'life cycle phases'.

In some studies, it is unclear if the sunk cost fallacy is committed regarding embodied impacts. This can be explained by the fact that the goal is often not unambiguously defined and the reasons for the inclusion of embodied impacts remain undocumented. In almost one third of studies, the choice to include or exclude landfilling is not explained, demonstrating what can be interpreted as a lack of concern for additional functions and/or transparency regarding methodological decisions. We recommend including landfilling in the system boundaries to the extent to which it is affected by the decision at hand and clearly reporting the underlying rationale.

**Material processing** steps such as comminution and separation are integral parts of mineral waste management. They are included in the system boundaries by the vast majority (96%) of studies. Material storage is addressed by 23% of studies. However, none of the studies that include storage identified it as a hotspot or a significant issue. Life cycle phases after the point of substitution are generally not included in the system boundaries, most likely because they are difficult to predict. This is especially true for mineral waste materials, as the application in road sub-bases and concrete indicate very long use phases. As discussed below, leaching during the use phase can be a significant issue, making this a significant research gap.

**Transportation** causes a potentially significant share of environmental impacts related to the recycling of low-value, high-density materials such as concrete, asphalt, and slag. Moreover, these materials are usually valorized regionally in the vicinity of their origin, making transport distances as well as means of transportation highly dependent on regional conditions. This is a potential cause of uncertainty when assessing the environmental feasibility of recycling options. Consequently, transport was identified as a significant issue by one third of studies under review, as well as by many studies that focus on waste consuming systems such as concrete production or road construction (not within the scope if this review), e.g., [40,43–50]. For example, Anastasiou et al. [45], Mladenovič et al. [51], and Turk et al. [47] found in multiple sensitivity analyses that the environmental feasibility of substituting natural aggregate with steel slag is dependent on transport distances, as steel slag aggregate has a higher density than natural aggregate and transport distances depend on local availability of both materials. They argue that, given a transport distance of 20 km for natural aggregate, the transport distance for steel slag must be below 37.4 [45] or 70 km [51] in order to lead to greenhouse gas savings. Assuming the same transport distance for both materials, steel slag utilization reduces greenhouse gas emissions for distances below 145 km [45] resp. 160 km [51]. This break-even point varies significantly between impact categories, meaning that recycling can be never (abiotic depletion) or always (human toxicity) environmentally feasible [51]. Turk et al. [47] determine that replacing natural aggregate with recycled aggregate is feasible with an additional transport distance of up to 100 km with respect to climate change, and likewise find a strong variation for other impact categories. Blengini and Garbarino [40] conclude that additional transport requirements outweigh the benefits of recycling if the transport distance of recycled aggregate exceeds that of natural aggregate by a factor of two to three. They further note that delivery distances for recycled aggregate depend on several factors, i.e., size of the recycling plant, plant location in mountains, or plain areas and the regional road network [40]. In a case study for Belgrade, Serbia, Marinković et al. [44] further identify the means of transportation as a significant parameter, assuming that recycled aggregate is transported by lorry, whereas natural aggregate is transported by ship. In total, the literature suggests that transport data are highly case-specific and a potential significant issue for LCAs of bulk mineral waste management systems.

Eighty-nine percent of studies under review include transportation in the foreground system. Transport is reported as a hotspot by 46% and as a significant issue by 33% of all studies, while none of the studies explicitly describe it as insignificant. We therefore recommend using scenario-specific data regarding both transport distances and means of transport. We further recommend conducting sensitivity analyses on transport distances, in the manner demonstrated by, e.g., [10,40,52–54].

**Leaching** of heavy metals is another potential significant issue, as 'non-hazardous' is often defined as the concentration or the leaching potential being below the legal threshold for the intended application. This does not mean that the aggregate is free of toxic elements such as heavy metals. The issue of leaching proves to be controversial in LCA literature on mineral waste management and mineral secondary material use (the latter being outside of the scope of this review). Faleschini et al. [55] conclude that leaching from electric arc furnace slag in concrete is below the threshold of Italian standards and in fact lower than that of natural aggregates. Chebbi et al. [56] found that while leaching from raw electric arc furnace slag is not within the legal limit in France, processed electric arc furnace slag does not exceed the threshold and can therefore be used in road construction. Mroueh et al. [43] determine that in their case study, the use of blast furnace slag in road construction is less polluting than the use of primary raw materials. Marion et al. [57] find that adding blast furnace slag does not affect the leaching behavior of concrete, keeping the measured concentrations of leached heavy metals well below the limits stated by EU Directive 98/83/EC and roughly the same as unpolluted Belgian soil. Chand et al. [58] consider the leaching behavior of converter slag as safe for landfill and for use as a building material. The results of Schwab et al. [59] contradict the above-mentioned studies and show that of the investigated materials, iron and steel slag leaches the largest amount of heavy metals, especially vanadium. Contaminated material poses a long-term threat to the environment. Depending on the type of soil, pollutants can be retained and may still contaminate groundwater after more than a hundred years, making leaching difficult to consider in LCA [59]. Butera et al. [46] conclude that for LCA of CDW management options, leaching creates significant impacts, making landfilling the preferable option for the categories human toxicity, ecosystem toxicity, and freshwater eutrophication.

As demonstrated by Marion et al. [57], there is no systematic relationship between the total heavy metals in cement and the proportion that leaches out. It is therefore not possible to draw conclusions about the elution behavior from total concentrations [57,60]. In addition to source concentrations, the leaching and fate of heavy metals are affected by several factors such as environmental conditions, condition of the structure and its surface [43], material properties, soil type [59], and pH [57], which is why the results of individual leaching tests cannot be generalized [43,59] and leaching tests at laboratory scale cannot be transferred to field conditions [57,61]. Further, heavy metal concentrations and their mineral bonds differ even across slag types [62], which is why results cannot simply be transferred to other material types. Due to the wide range of factors affecting leaching, Schwab et al. [59] advise the use of case-specific data. Results from the Environment Agency Austria [62] conclude that vanadium and molybdenum could leach in harmful concentrations if converter slag is used in unbound applications, while significantly lower concentrations are to be expected in bound road construction. This shows that the type of application of the mineral waste material matters in regard to its leaching potential.

There is a wide range of heavy metal compositions in natural aggregates [62]; therefore, natural aggregates can leach heavy metals as well. This means that leaching needs to be considered for both primary and secondary materials. Further, leaching of waste materials can occur when recycled as aggregate and when landfilled. When conducting LCA for decision support, it is crucial to consider how the leaching behavior of all material flows changes as a consequence of the decision at hand.

Leaching is often overlooked, even in LCA studies focusing on systems consuming bulk mineral waste materials such as concrete production or road construction (which are not within the scope of this review). Although a few studies include the issue [43,55,63], the vast majority omit it, e.g., [45,47,51,64–68]. This is often due to the fact that leaching data are not available [61]. Of the body of literature assessed in this review, just four studies (7%) include leaching data. The choice to address or not to address leaching is explained by six studies in total. Butera et al. [46] performed the only study that uses scenario-specific leaching data. Three studies take leaching from the secondary material in its application into account. Leaching from the displaced primary material and potentially

avoided leaching of the landfilled waste material are included by two and three studies, respectively.

We argue that LCA studies on bulk mineral waste management should include leaching data if possible, or explain why they do not. LCA studies not referring to specifically defined waste materials and/or use cases due to a broader scope cannot include scenario-specific leaching data. This limitation should be acknowledged by LCA practitioners and critically discussed regarding the applicability of LCA results in specific improvement contexts. As Laurent et al. [26] explain for the case of landfills, failing to include leaching data means modeling a "free sink for pollution". We refer to Laurent et al. [26], Schwab et al. [59], and Butera et al. [46] for further recommendations on addressing leaching in LCA of mineral waste management.

### 3.2. Inventory Analysis

Transparency and reproducibility are essential requirements for the credibility of any scientific study. Regarding the LCI phase, this means providing both foreground and background inventory data (unless confidential) and clearly documenting and justifying data choices. A third-party report should include qualitative and quantitative descriptions of unit processes as well as sources of the respective inventory data [12]. Therefore, foreground inventory data as well as background database, database version, and used datasets need to be disclosed. As of ecoinvent version 3.0 [69], three system models are available: two for attributional modeling and one for consequential modeling. Between the system models, life cycle inventories and impact assessment results for the same foreground system can deviate significantly, even when using the same database version [38]. If multiple system models are available for the chosen background database, we consider it essential to report and justify the choice.

Our assessment of inventory documentation in the considered studies is summarized in Figure 8. The foreground inventory is documented by 68% of studies. Note that we did not assess whether the system is documented in "sufficient detail" [12] to enable full reproduction of the respective foreground systems. The background database, the database version, and the used datasets are recorded by 70%, 46%, and 26%, respectively (four studies—7%—did not use a background database). Out of the 19 studies that use ecoinvent v3.0 or later, just six report the background system model and two justify this choice.

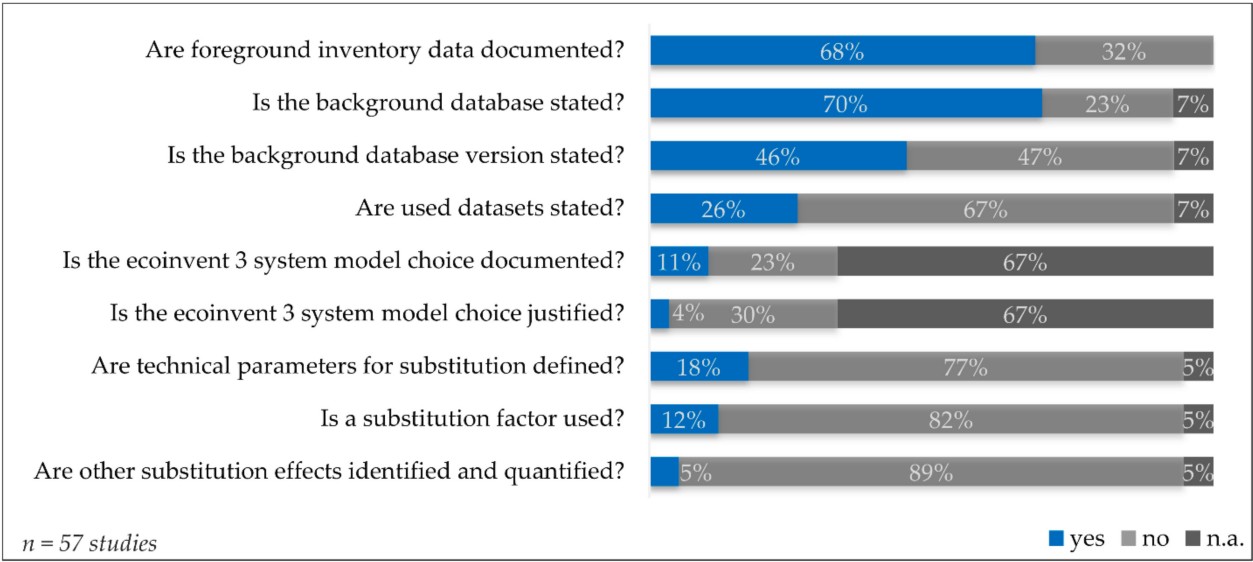

**Figure 8.** Assessment results for the category 'inventory analysis' (n.a. = not applicable: 7% did not use a background database; 67% did not use ecoinvent 3; 5% did not consider substitution).

The assessment results expose a lack of transparency with respect to inventory data across the body of literature. The fact that few of the studies that use ecoinvent 3.0 or later address the chosen system model may further indicate a lack of understanding of the system models available in the ecoinvent database. We recommend using the system model that most accurately represents the modeling approach of the foreground system in line with the goal and scope of the study. We appeal to LCA practitioners to accurately report the database, the database version, the used datasets, and, if applicable, the background system model.

Reuse and recycling can affect the properties of materials [12]. As the physical and chemical properties of recycled concrete and slag deviate from those of natural aggregates, it may not be correct to assume a 1:1 substitution. For example, due to the angular grain shape and the resulting larger surface area of recycled aggregates, their application in concrete mixes may change the total demand for binder [70–74] and consequently the required amount of water or superplasticizer [75]. Recycled fine aggregates contain cement residues, which increase the water absorption capacity and in turn increase the water/cement ratio and decrease the workability and final mechanical performance of the concrete [76]. Due to the energy intensive calcination process, the potential increase in binder consumption can have significant environmental implications, which may outweigh the benefits of using recycled aggregate in concrete [8,44]. In contrast, partial replacement of primary aggregate with recycled aggregate does not significantly change the compressive strength of concrete and therefore does not increase in the required amount of binder [77]. Reclaimed asphalt pavement can partly substitute bitumen in addition to aggregate if it is recycled into new asphalt concrete. Yet, the physical and chemical properties of old binder can affect, e.g., the service life of the new asphalt pavement [78]. LCA studies focusing on the production of cement or concrete (which are not within the scope of this review) often include binding equivalent or compressive strength in the functional unit [47,79–81]. Panesar et al. [82] investigate the impact of the choice and complexity of the functional unit on LCA results for concrete production and conclude that it should include all relevant technical parameters and expected exposition. Similarly, van den Heede and de Belie [18] recommend including parameters of functional performance such as strength, durability, and lifespan in the functional unit of concrete production. This shows that the intended application of the waste material and the associated technical parameters can not only affect the substitution factor, but may cause second-order substitution effects. Hence, these factors are relevant for LCA studies on mineral waste management.

Technical parameters affecting substitution are identified by 18% of studies. Commonly identified parameters are compressive strength, binding equivalent, and workability of the product (for cement and concrete applications), density, and durability. Twelve percent of studies employ a substitution factor (including 1:1) to account for differences in technical properties between primary and secondary materials. Three studies (5%) account for second-order substitution effects resulting from different technical properties. The majority of studies either use a 1:1 substitution or do not document the substitution factor. Interestingly, only three studies identified material properties previously defined for the waste material (see Section 3.1.2) as significant for substitution. Two of these studies based the substitution factor on the material property of the waste compared to the competing virgin material. Two studies identified substitution effects based on the properties of the product containing recycled material (concrete and asphalt, respectively).

If the secondary aggregate is intended to substitute primary aggregate such as natural gravel or crushed limestone in bound applications like concrete or asphalt wearing courses, we recommend clearly documenting all assumptions regarding changes in demand for binder, water, and plasticizer, as well as changes in compressive strength, abrasion resistance, lifetime, and other properties of the concrete or asphalt. For best practice examples, we point to most LCA studies assessing common use cases for secondary aggregates and supplementary cementitious materials (both outside of the scope of this review), such as concrete production, e.g., [44,70,75,83], or road construction, e.g., [51].

### 3.3. Impact Assessment

According to ISO 14040/44 [11,12], the selection of impact categories and characterization models has to be reported and justified with respect to the goal of the study. ISO 14044 further recommends that weighting should not be employed in comparative LCA studies intended for publication [12]. Because weighting impact categories is inherently normative, it is important for decision-makers to have unweighted results available, unless it is clear that the weighting is done according to the values represented by the decision-maker. For transparency reasons, it is in fact irremissible to present unweighted LCIA results, so that the weighting process and subsequent decision-making can be comprehensible for potential stakeholders. LCA considers "all attributes or aspects of natural environment, human health, and resources" [11]. This is necessary to identify potential problem shifting across media or category endpoints. We acknowledge that it can make sense to exclude impact categories from the assessment. Due to different value systems and potentially low relative impacts, not all impact categories are relevant for decision-makers in every decision-making context. Nevertheless, it is crucial to report the reasons for the exclusion of impact categories, so that misinterpretation of the results can be avoided.

As shown in Figure 9, LCIA methods and unweighted LCIA results are documented in the majority of studies. The choice of LCIA methods is reported by 81% and justified by 14% of studies. While 89% of studies either remain at midpoint or present LCIA results at both mid- and endpoint level, 11% do not report unweighted impact assessment results, presenting only weighted endpoint indicators. Sixteen percent explain why specific midpoint impact categories (not counting single-score results or aggregated endpoint categories such as human health) are excluded from their assessment. Figure 10 shows how many studies justify the exclusion of impact categories by number of impact categories covered. More than half of all studies cover between zero and five impact categories, while 23% of studies investigate 12 categories or more.

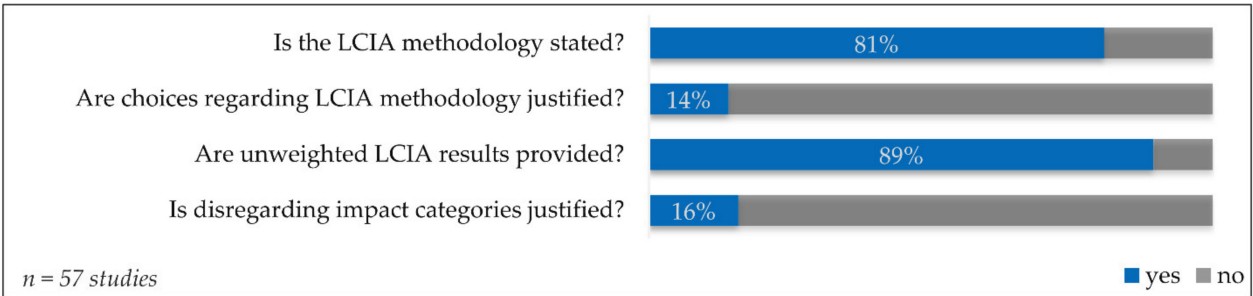

**Figure 9.** Assessment results for the category 'impact assessment'.

There is room for more transparency regarding choices for and against certain characterization models and impact categories. A positive example regarding the latter is Borghi et al. [10], who address 15 impact categories and explain why additional categories are excluded. We urge LCA practitioners to transparently document assumptions and value choices regarding these aspects, as they may affect the overall conclusions of an LCA and can introduce subjectivity into the LCIA phase [11]. These data need to be reported for robust decision support.

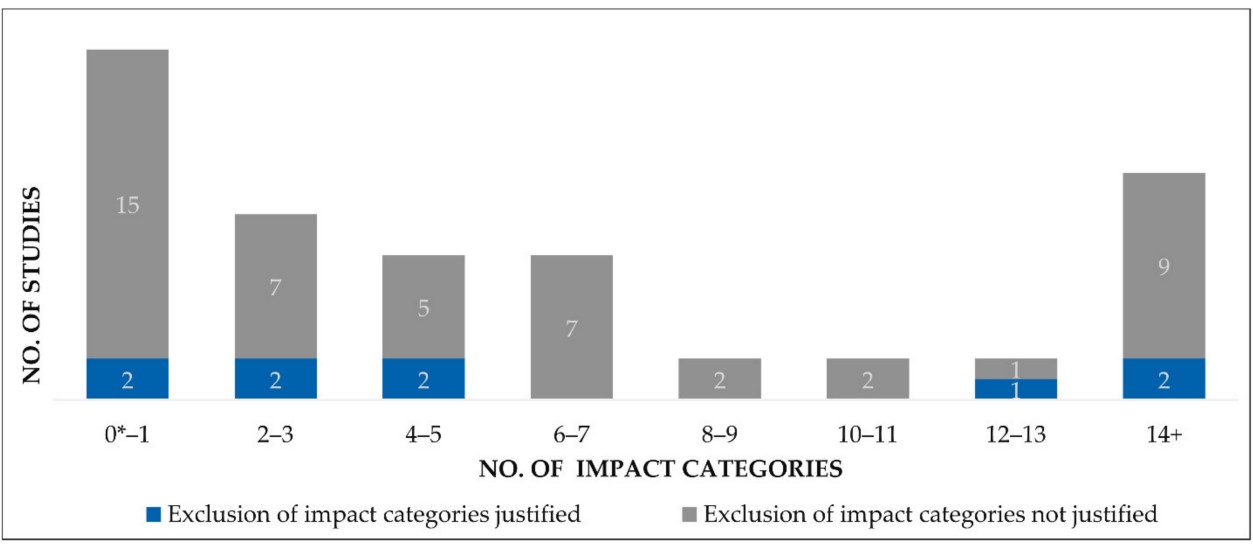

**Figure 10.** Number of studies justifying the exclusion of impact categories by number of midpoint impact categories covered (* two studies did not report any midpoint impact categories).

### 3.4. Interpretation

The interpretation phase helps to understand and evaluate LCI and LCIA results in the context of significant issues and possible limitations. To enable robust conclusions and recommendations, it is necessary that interpretation happens in line with the goal and scope of the LCA and that conclusions and recommendations take the evaluation element into account [11]. To this end, ISO 14044 calls for sensitivity analyses in comparative LCA studies intended for publication and recommends performing completeness, sensitivity, and consistency checks [12]. ISO 14044 requires detailed and transparent reporting of limitations and further specifies that assumptions and limitations regarding data and methodology that may affect the interpretation phase need to be discussed [12].

The assessment results for the interpretation phase are visualized in Figure 11: 49% of studies report sensitivity analyses and 44% identify significant issues. Sensitivity, completeness, and consistency checks were found in 32%, 16%, and 19%, respectively. Several studies identify, e.g., data gaps. However, few discuss if and how this affects the conclusions. The terms 'sensitivity check', 'completeness check', and 'consistency check' are not commonly used in the literature, which means that these criteria had to be understood from the context through manual screening and interpretation (see Table S3 in the Supplementary Materials for the conditions to be met for each criterion). Although limitations are identified in 47% of the reviewed literature, their discussion is usually narrow. Few studies present comprehensive discussions of the limitations regarding methodology and available data (e.g., [16,84]).

This can be interpreted as a lack of critical analysis and reflection of LCA results within the body of literature. In order to support robust decision-making, a good understanding and documentation of limitations and significant issues is imperative. A minimum documentation in the context of bulk mineral waste management systems should include sensitivity analyses regarding transport distances as demonstrated by, e.g., [10,40,52–54]. The sensitivity analysis results should be discussed within a sensitivity check before drawing conclusions and providing recommendations to decision-makers. We further recommend clearly reporting known limitations regarding, e.g., data uncertainty, consistency, and completeness and discussing them with respect to conclusions and recommendations.

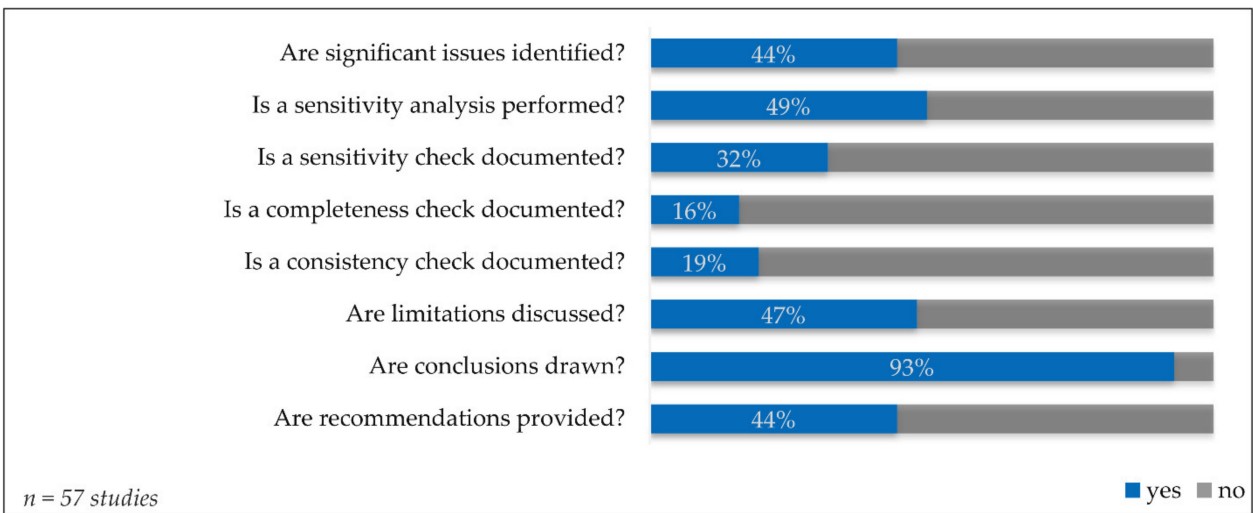

**Figure 11.** Assessment results for the category 'interpretation'.

Drawing conclusions is a requirement defined in the ISO standards [11,12] and is necessary to understand the meaning of the LCIA results in light of present limitations. We argue that the direct application of any LCA is always a decision-making context and hence the goal is always decision support. Therefore, giving "specific recommendations to decision-makers" [12] is always appropriate to the goal and scope. Recommendations, which should "reflect a logical and reasonable consequence of the conclusions" and "relate to the intended application" [12], close the loop to the decision to be supported, which is identified during the goal definition.

Conclusions are documented in 93% of studies, but recommendations are expressed by fewer than half (44%). Interestingly, the latter share is higher than the 37% of studies that name decision support as a goal and the 32% that identify decisions to be supported. This can be explained by the fact that recommendations are sometimes not clearly derived from the LCIA results, limitations, and conclusions.

These results could lead to the interpretation that decision support is in some cases an afterthought rather than a goal. Few studies derive clear and specific recommendations regarding the identified decision-making context and address them to the identified decision-makers. A good example can be found in Pantini et al. [5]. We strongly recommend basing recommendations on the specific decision-making context identified and considering the type of recommendations required for robust decision-making during the goal definition phase. This enables practitioners to design all parts of the LCA accordingly. Lastly, we recommend clearly documenting when results are inconclusive and reflecting this fact in the conclusions and recommendations. As Yang [34] accurately points out, "inconclusiveness should be commonplace" in LCA, because complex systems with inherently high uncertainty are analyzed.

### 4. Limitations

Several limitations of this review need to be discussed. This review aims to identify central shortcomings regarding decision support in the context of bulk mineral waste management and does not represent an exhaustive assessment of LCA practice in general. Moreover, all results and recommendations are based on the body of literature as a whole and are not intended to criticize individual studies. We decided not to provide detailed assessment results for each study, as this would mean singling out studies. Even if the studies are anonymized, it would likely be possible to identify studies based on their scores. We therefore published only aggregated results in form of percentages of the body of literature for each criterion.

The selection of assessment criteria is non-exhaustive, focusing on criteria that (a) were regarded as crucial for decision support and (b) could feasibly be assessed in a yes-no

question scheme. Some important criteria, e.g., whether the system is documented in "sufficient detail" [12] to enable full reproduction of the respective foreground systems, could not feasibly be assessed and were excluded.

Although the literature search and study selection were conducted in a structured way and were repeated once for verification, this was only done by the first author. Conducting the data extraction for the quantitative assessment posed varying degrees of difficulty. Some criteria could be validated by a keyword search (e.g., functional unit), while others relied heavily on manual screening and interpretation (e.g., foreground inventory data, completeness, and consistency checks; for more information, see Table S3 in the Supplementary Materials). The effort was exacerbated by studies that deviated from the terminology used in the ISO standards. Therefore, the data extraction approach does not guarantee absolute accuracy and must be considered an inherent limitation of this review. Note that it was generally ruled in favor of an article if a criterion could be understood from the context, even if it was not unambiguously documented.

The quantitative assessment results represent a binary assessment of the literature. This approach does not paint the full picture, because (binary) quantitative assessment results do not necessarily reflect quality. For this reason, we supplemented the quantitative assessment with a qualitative assessment where necessary.

## 5. Conclusions and Recommendations

In this article, a detailed analysis of methodological shortcomings of the screened literature on LCA of bulk mineral waste management systems was conducted. The aim of this review was not only to analyze methodological issues of LCA applied in a specific sector, but to select and analyze issues that are linked to decision support. With this special emphasis on suitability for decision support, we explore a new and relevant research aspect and present a new synthesis of the analyzed literature. Are the existing LCA case studies on bulk mineral waste management suitable for decision support? In this regard, the results show widespread inadequacies regarding methodology and documentation in all LCA phases, i.e., goal and scope definition, inventory analysis, impact assessment, and interpretation. The issues we consider most important are listed below with recommendations for future LCA practice. The issues are categorized into two groups: those applicable to LCA methodology in general and those applicable to LCA of bulk mineral waste management systems. We refer to Laurent et al. [26] for additional recommendations for good practice in the broader scope of LCA applied to solid waste management systems.

**LCA in General**:

- Goal definition: clearly disclose to what end the LCA is performed and consider the decision-making context in which the LCA results are likely to be applied in practice. The goal definition impacts all subsequent LCA phases, including scope definition, LCI, and LCIA.
- Functional unit: include the main functions of the systems in the functional unit. A functional unit without a function is not a functional unit.
- System boundaries/multifunctionality: design your model to represent the effects of all considered options (and the counterfactual, if applicable). Keep the cause–effect principle intact by using a consequential modeling approach, including the use of marginal data. This issue can be supported by supplementing LCA with additional methods such as material flow analysis, integrated assessment models, agent-based modeling, general and partial equilibrium models, and other approaches, as these may be more suitable for certain decision-making context situations than the linear modeling approach commonly applied in LCA [34].
- LCIA: report and justify your choice of impact categories and LCIA methods.
- Interpretation: critically assess your results regarding consistency, completeness, and sensitivity. Transparently report and discuss limitations. Draw conclusions considering these limitations. Give recommendations to decision-makers regarding

the decisions identified in the goal definition phase. Highlight inconclusiveness if applicable.

**LCA of Bulk Mineral Waste Management:**

- Life cycle phases: discuss whether embodied impacts of the waste materials and avoided landfilling are affected by the decision at hand. Include them in the system boundaries to the degree they are affected.
- Transport: Use case-specific data regarding transport distances and means of transport. Conduct a sensitivity analysis regarding transport distances.
- Leaching: include leaching data or clearly explain why they are omitted. For decision support, scenario-specific leaching data should represent the changes between different courses of action and/or the counterfactual.
- Substitution effects: account for substitution effects (e.g., potentially increased demand for cement when substituting natural aggregate with secondary aggregate in concrete).

Understandably, certain details have to be omitted if studies are published in journals that place restrictions on the length of articles. The following generic example sentences demonstrate how all nine criteria regarding the goal definition (see Section 3.1.1) can be fulfilled efficiently: "This LCA addresses regional authorities and CDW recyclers in [region] and is intended to support their decision whether CDW in [region] should be utilized as recycled aggregate in road construction instead of being landfilled. This decision will affect all CDW in [region] for the years [years], a total estimated amount of [amount]." We recommend the extensive use of Supplementary Materials to report additional details on the decision-making context as well as on other criteria identified in this review (e.g., foreground inventory data, used datasets, waste material compositions, sensitivity analysis results). We consider reporting all relevant information crucial to prevent misinterpreting LCA results.

**Supplementary Materials:** The following are available online at https://www.mdpi.com/article/10.3390/su13094686/s1. Table S1: Completed PRISMA checklist, Table S2: Search stings, refinement, and search date for literature search in Web of Science and Scopus, Table S3: Assessment criteria, corresponding search terms and explanation.

**Author Contributions:** Conceptualization: C.D.; methodology: C.D. and A.C.; validation: T.H., V.Z. and W.B.; formal analysis: C.D.; investigation: C.D.; data curation: C.D.; writing—original draft preparation: C.D.; writing—review and editing: C.D., T.H., A.C., V.Z. and W.B.; visualization: C.D.; supervision: V.Z. All authors have read and agreed to the published version of the manuscript.

**Funding:** We acknowledge support by the Deutsche Forschungsgemeinschaft (DFG—German Research Foundation) and the Open Access Publishing Fund of Technical University of Darmstadt.

**Data Availability Statement:** The completed PRISMA-table, search strings, and refinement of the literature search as well as keywords and explanation of the assessment criteria are available in the Supplementary Materials. Further data are available upon request from the first author.

**Acknowledgments:** We thank our colleagues Benjamin Portner, Othman Mrani, and Almut Güldemund as well as the three anonymous reviewers for their valuable feedback.

**Conflicts of Interest:** The authors declare no conflict of interest.

## Appendix A

Table A1. List of assessed LCA studies on bulk mineral waste management (57 studies in total).

| 01 | Amato et al. [85] | Strategies of disaster waste management after an earthquake: A sustainability assessment |
|---|---|---|
| 02 | Basti [86] | Sustainable management of debris from the L'Aquila earthquake: environmental strategies and impact assessment |
| 03 | Bizcocho and Llatas [15] | Inclusion of prevention scenarios in LCA of construction waste management |
| 04 | Blengini [87] | Life cycle of buildings, demolition and recycling potential: A case study in Turin, Italy |
| 05 | Blengini and Garbarino [40] | Resources and waste management in Turin (Italy): the role of recycled aggregates in the sustainable supply mix |
| 06 | Borghi et al. [10] | Life cycle assessment of non-hazardous Construction and Demolition Waste (CDW) management in Lombardy Region (Italy) |
| 07 | Butera et al. [46] | Life cycle assessment of construction and demolition waste management |
| 08 | Chebbi et al. [56] | Environmental assessment of EAF slag in different "end of 2nd life" |
| 09 | Chen et al. [88] | Life Cycle Assessment of Internal Recycling Options of Steel Slag in Chinese Iron and Steel Industry |
| 10 | Coelho and de Brito [89] | Influence of construction and demolition waste management on the environmental impact of buildings |
| 11 | Dahlbo et al. [90] | Construction and demolition waste management—a holistic evaluation of environmental performance |
| 12 | Di Maria et al. [16] | Downcycling versus recycling of construction and demolition waste: Combining LCA and LCC to support sustainable policy making |
| 13 | Dong et al. [91] | Achieving carbon emission reduction through industrial & urban symbiosis |
| 14 | Faleschini et al. [92] | Sustainable management of demolition waste in post-quake recovery processes: The Italian experience |
| 15 | Fort and Cerny [93] | Transition to circular economy in the construction industry: Environmental aspects of waste brick recycling scenarios |
| 16 | Guignot et al. [52] | Recycling Construction and Demolition Wastes as Building Materials: A Life Cycle Assessment |
| 17 | Hossain and Ng [94] | Influence of waste materials on buildings' life cycle environmental impacts: Adopting resource recovery principle |
| 18 | Jain et al. [54] | Environmental life cycle assessment of construction and demolition waste recycling: A case of urban India |
| 19 | Karanović et al. [95] | Assessment of construction and demolition waste management in the city of Aveiro, Portugal |
| 20 | Klang et al. [96] | Sustainable management of demolition waste—an integrated model for the evaluation of environmental, economic and social aspects |
| 21 | Kua [97] | The Consequences of Substituting Sand with Used Copper Slag in Construction |
| 22 | Kucukvar et al. [98] | Life Cycle Assessment and Optimization-Based Decision Analysis of Construction Waste Recycling for a LEED-Certified University Building |
| 23 | Lee and Park [99] | Estimation of the environmental credit for the recycling of granulated blast furnace slag based on LCA |
| 24 | Levis et al. [100] | Quantifying the Greenhouse Gas Emission Reductions Associated with Recycling Hot Mix Asphalt |
| 25 | Li et al. [101] | Environmental impact assessment of mobile recycling of demolition waste in Shenzhen, China |
| 26 | Liu et al. [102] | Economic and Environmental Assessment of Carbon Emissions from Demolition Waste Based on LCA and LCC |

**Table A1.** *Cont.*

| 27 | Lockrey et al. [84] | Concrete recycling life cycle flows and performance from construction and demolition waste in Hanoi |
|---|---|---|
| 28 | Mah et al. [103] | Life cycle assessment and life cycle costing toward eco-efficiency concrete waste management in Malaysia |
| 29 | Mah et al. [104] | Environmental impacts of construction and demolition waste management alternatives |
| 30 | Mah et al. [105] | Concrete waste management decision analysis based on life cycle assessment |
| 31 | Martínez et al. [41] | End of life of buildings: three alternatives, two scenarios. A case study |
| 32 | Mastrucci et al. [106] | Geospatial characterization of building material stocks for the life cycle assessment of end-of-life scenarios at the urban scale |
| 33 | Mercante et al. [107] | Life cycle assessment of construction and demolition waste management systems: a Spanish case study |
| 34 | Miliutenko et al. [78] | Opportunities for environmentally improved asphalt recycling: the example of Sweden |
| 35 | Mousavi et al. [108] | Decision-based territorial life cycle assessment for the management of cement concrete demolition waste |
| 36 | Ortiz et al. [109] | Environmental performance of construction waste: comparing three scenarios from a case study in Catalonia, Spain |
| 37 | Pantini et al. [5] | Towards resource-efficient management of asphalt waste in Lombardy region (Italy): Identification of effective strategies based on the LCA methodology |
| 38 | Penteado and Rosado [49] | Comparison of scenarios for the integrated management of construction and demolition waste by life cycle assessment: A case study in Brazil |
| 39 | Ram et al. [110] | Environmental benefits of construction and demolition debris recycling: Evidence from an Indian case study using life cycle assessment |
| 40 | Rosado et al. [42] | Life cycle assessment of construction and demolition waste management in a large area of São Paulo State, Brazil |
| 41 | Simion et al. [111] | Comparing environmental impacts of natural inert and recycled construction and demolition waste processing using LCA |
| 42 | Simion et al. [112] | Ecological footprint applied in the assessment of construction and demolition waste integrated management |
| 43 | Song et al. [113] | Exploring the life cycle management of industrial solid waste in the case of copper slag |
| 44 | Vieira and Horvath [114] | Assessing the end-of-life impacts of buildings |
| 45 | Vitale et al. [115] | Life cycle assessment of the end-of-life phase of a residential building |
| 46 | Vossberg et al. [53] | An energetic life cycle assessment of C&D waste and container glass recycling in Cape Town, South Africa |
| 47 | Wang et al. [116] | Considering life-cycle environmental impacts and society's willingness for optimizing construction and demolition waste management fee: An empirical study of China |
| 48 | Wang et al. [117] | Combining life cycle assessment and Building Information Modelling to account for carbon emission of building demolition waste: A case study |
| 49 | Wang et al. [118] | Estimating the environmental costs and benefits of demolition waste using life cycle assessment and willingness-to-pay: A case study in Shenzhen |
| 50 | Wang et al. [119] | Energy–environment–economy evaluations of commercial scale systems for blast furnace slag treatment: Dry slag granulation vs. water quenching |
| 51 | Wu et al. [120] | Quantification of carbon emission of construction waste by using streamlined LCA: a case study of Shenzhen, China |
| 52 | Yahya and Boussabaine [121] | Quantifying environmental impacts and eco-costs from brick waste |
| 53 | Yazdanbakhsh [77] | A bi-level environmental impact assessment framework for comparing construction and demolition waste management strategies |

**Table A1.** *Cont.*

| 54 | Zambrana-Vasquez et al. [122] | Analysis of the environmental performance of life-cycle building waste management strategies in tertiary buildings |
|---|---|---|
| 55 | Zeng et al. [123] | Greenhouse gases emissions from solid waste: an analysis of Expo 2010 Shanghai, China |
| 56 | Zhang et al. [124] | Eco-efficiency assessment of technological innovations in high-grade concrete recycling |
| 57 | Zhang et al. [125] | Co-benefits of urban concrete recycling on the mitigation of greenhouse gas emissions and land use change: A case in Chongqing metropolis, China |

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
