# Peer review of "Are LCA Studies on Bulk Mineral Waste Management Suitable for Decision Support? A Critical Review"

_sustainability, doi:10.3390/su13094686_

Round 1

Reviewer 1 Report

The article addresses an interesting topic, consistent with the journal’s aim.

The paper is an extensive review on several works about the LCA of bulk mineral waste. In this sense, maybe the title is too ambitious. Indeed, the authors do not provide a real answer to the to the question they ask in the title.

The majority of the limitations and problems they encounter are typical of LCA in general (substitution, allocation, functional unit, landfill avoiding etc), and not so much of the particular sector they have studied. The author should emphasize specific issues more. For example, I believe that transport and landfill have a great impact in a particular sector like this (in which the bulk has a relevant specific weight). It would be interesting to show the impact these categories have with respect to the others, in order to quantify how much one choice can affect another (whether or not to consider these processes).

Almost all outputs are yes/no. It would be interesting to quantify more the impacts of the categories for each work (e.g. the percentile of the category) to understand where it is most important to focus attention.

More or less, all the references to the captions of the figures in the main text are wrong.

Lastly I advice to stress the conclusion on the aim that the authors that the authors have given themselves, to avoid it being just a photograph of the current state.

Author Response

Dear Reviewer,

thank you very much for your comments! We attached the manuscript with "track changes" enabled for you to see any changes we made. Please find below our point-by-point response (our response in red):

Point 1: The paper is an extensive review on several works about the LCA of bulk mineral waste. In this sense, maybe the title is too ambitious. Indeed, the authors do not provide a real answer to the to the question they ask in the title.

Response 1: We believe that we did, in fact answer the titular research question adequately within the “Conclusion” section. However, we agree that we avoided a direct answer and that this polite way of answering may not be understood as such by the reader. We therefore added the research question directly ahead of our answer to point out more clearly that it is, in fact, our answer to the research question (line 757ff):

“Are the existing LCA case studies on bulk mineral waste management suitable for decision support? In this regard, the results show widespread inadequacies regarding methodology and documentation in all LCA phases […]”

We hope that this minor but important change makes our answer clear.

Point 2: The majority of the limitations and problems they encounter are typical of LCA in general (substitution, allocation, functional unit, landfill avoiding etc), and not so much of the particular sector they have studied. The author should emphasize specific issues more. For example, I believe that transport and landfill have a great impact in a particular sector like this (in which the bulk has a relevant specific weight).

Response 2: We agree that specific issues are important in a review such as ours. For this reason, we emphasized several specific issues. Especially transport (lines 447-483) and leaching (lines 484-547) were discussed in detail in section 3.1.4. Bulk density has been addressed as relevant for transportation in section 3.1.2. We briefly discuss leaching from landfills within our discussion on leaching in section 3.1.4 (lines 484-547). Landfilling (or rather avoided landfilling), however, is only significant if it is a feasible option within the specific decision context. In countries or regions with already high recovery rates, landfilling may not be affected at all – instead, the focus of an LCA could lie on the change from one valorization route to another. Consequently, our focus in regard to landfilling lies on the decision whether or not to include it in the system boundaries.

We have added some references on the potential significance of landfilling in lines 415ff.

Point 3: It would be interesting to show the impact these categories have with respect to the others, in order to quantify how much one choice can affect another (whether or not to consider these processes).

Response 3: We considered quantifying (1) the importance of the criteria with respect to each other and (2) their interconnection, but concluded that it is not feasible. (1) Quantifying the importance would require the development of a weighting scheme for the criteria which would be highly uncertain and subjective (similar to weighting in LCIA). This is further exacerbated by the fact that different criteria can affect LCA results to different degrees depending on the specific study. (2) The interconnection between criteria also differs between studies and may only be somewhat quantifiable for a handful of criteria combinations. Your example of bulk density (criterion “Are technical properties of the waste material defined?”) affecting the impact of transportation (criterion: “Is transport included?”) would probably be the “easiest” to conduct. However, it still depends at the very least on transport distances and means of transportation of all transportation routes within the foreground system. This would require the development of a calculation model for each combination of criteria to be investigated. As this type of investigation is clearly outside of the scope of a critical review, we decided against including it in our review.

We did, however, point out a number of such interconnections, e.g.: technical properties are relevant for transportation and leaching (section 3.1.2, lines 309-313); the recommendations depend on – and should reflect – the goal of the study (section 3.4, lines 700-719).

Point 4: Almost all outputs are yes/no. It would be interesting to quantify more the impacts of the categories for each work (e.g. the percentile of the category) to understand where it is most important to focus attention.

Response 4: In the conceptual phase of the article we did consider analyzing the results for (1) categories as a whole and (2) single (unaggregated) studies, but decided against this idea. Similar to Point 3, the former (1) would require developing a (subjective and uncertain) weighting scheme for the criteria. Note that equal weighting (1:1:1…) is still subjective weighting. We therefore doubt that any robust conclusions could be drawn from such an analysis. The latter (2) would require providing detailed assessment results for each study – thereby singling out studies. Even if the studies were anonymized, it would likely be possible to identify studies based on their scores.

We have added this fact to the limitations section (section 4, lines 725-729).

Point 5: More or less, all the references to the captions of the figures in the main text are wrong.

Response 5: The links were disabled during editing at MDPI. We have contacted MDPI about this issue and removed all links to figures, tables and references according to MDPI’s advice.

Point 6: Lastly I advice to stress the conclusion on the aim that the authors that the authors have given themselves, to avoid it being just a photograph of the current state.

Response 6:  Thank you for this comment! In addition to the change mentioned in Response 1, we added the following passage in the beginning of the conclusion (lines 752-756):

“The aim of this review was not only to analyze methodological issues of LCA applied in a specific sector, but to select and analyze issues that are linked to decision support. With this special emphasis on suitability for decision support we explore a new and relevant research aspect and present a new synthesis of the analyzed literature.”

We believe that the addition of this passage emphasizes the fact that our critical analysis is more than just a photograph of the current state.

Reviewer 2 Report

The reviewer recognizes that the manuscript provides important information and analyzed insight how the LCA studies were applied and used by relevant stakeholders for decision making how to manage bulk mineral waste in environmentally sound manner. The reviewer expects that this manuscript would become important information source for experts and officials to deepen their expertise and knowledge on bulk mineral waste management.

The reviewer has 2 comments:

1) There are many technical errors of links reference names and numbers. Please revise them;

2) Whether or not it is possible to further discuss results of LCA at post-closure long-term phase. The current manuscript seems to only discuss operational phase of bulk mineral waste; but the reviewer thinks that there are some references analyzing long-term LCA.

Author Response

Dear Reviewer,

thank you very much for your comments! We attached the manuscript with "track changes" enabled for you to see any changes we made. Please find below our point-by-point response (our response in red):

Point 1: There are many technical errors of links reference names and numbers. Please revise them;

Response 1: The links were disabled during editing at MDPI. We have contacted MDPI about this issue and removed all links to figures, tables and references according to MDPI’s advice.

Point 2: Whether or not it is possible to further discuss results of LCA at post-closure long-term phase. The current manuscript seems to only discuss operational phase of bulk mineral waste; but the reviewer thinks that there are some references analyzing long-term LCA.

Response 2: Long-term effects are absolutely an interesting and important topic. Unfortunately, this is only marginally addressed in current literature. We criticize this in section 3.1.4. (lines 441-446):

“Life cycle phases after the point of substitution are generally not included in the system boundaries, most likely because they are difficult to predict. This is especially true for mineral waste materials, as the application in road sub-bases and concrete indicate very long use phases. As discussed below, leaching during the use phase can be a significant issue, making this a significant research gap.”

Within the same section, we discuss leaching in particular (lines 484-547), including the difficulties in addressing it within the framework of LCA. Leaching is, to the best of our knowledge, the most significant (potential) long-term effect during the use phase of recycled and landfilled mineral waste materials.

Note that the scope of our review includes only studies focusing on systems for the treatment of mineral waste, not systems consuming it (e.g., “green” concrete production, road construction). For the latter, there are indeed several studies that include later life cycle phases such as use phase of the recycled material or second end-of-life (e.g., 10.1016/j.resconrec.2009.08.011; 10.1007/s11367-017-1284-0; 10.1007/s11367-013-0614-0; 10.1016/j.resconrec.2015.05.009). These studies are, however, not within the scope of our review. This selection criterion is explained in lines 158-165.

Reviewer 3 Report

The study provides a detailed Review of the state - of - art in the field of LCA analyses in bulk mineral waste management. However, the supplementary tables provided by the authors are not accessible, thus a detailed review of the references managed could not be accomplished. The study analyzes in-depth all aspects with regards to the application of LCA methodology, but an overview of the papers considered should have been made in order to check possible relations among criteria or the percentage of criteria satisfied in each study. Some detailed comments about the paper can be found in the attached document. 

Author Response

Dear Reviewer,

thank you very much for your comments! We attached the manuscript with "track changes" enabled for you to see any changes we made. the document also contains the comments you provided in the PDF file as well as our replies. Please find below our point-by-point response (our response in red):

Point 1: However, the supplementary tables provided by the authors are not accessible, thus a detailed review of the references managed could not be accomplished.

The link in the document is not functional yet, as the article is not yet published. We have contacted MDPI about that fact that the supplementary material was not accessible to you. There may have been a problem with the upload. We received the following reply to this point:

“[…] When you submit revised paper, there will be an button to uploading supplementary. Please upload the materials in Word or PDF version. You also can send me the supplementary by email. I will help upload.

The link in document will update when paper is published. Before that the link will not work and reviewers will need download the supplementary from our system. I can help send supplementary to reviewers by email if you do not mind. […]”

We will re-upload the supplementary material during re-submission and, in addition, will ask the Assigned Editor to provide it to you by Email (just in case there is a problem with the upload).

Point 2: The study analyzes in-depth all aspects with regards to the application of LCA methodology, but an overview of the papers considered should have been made in order to check possible relations among criteria or the percentage of criteria satisfied in each study.

Response 2: We considered presenting detailed assessment results for each reviewed paper, but concluded that this would mean singling out studies. Even if the studies were anonymized, it would likely be possible to identify studies based on their scores. We have added the fact that detailed assessment results for each paper cannot be published to the limitations section (section 4, lines 725-729). The data availability statement (lines 822-824) reads: “[…] Further data are available upon request from the first author.” This means that we will gladly provide authors of reviewed papers with the results to their respective papers.

Point 3: Some detailed comments about the paper can be found in the attached document.

Response 3: Please find our changes and replies to these comments within the attached manuscript.

Round 2

Reviewer 1 Report

The authors addressed my requests